# Isolation, Characterization, and Application of a Bacteriophage Infecting the Fish Pathogen *Aeromonas hydrophila*

**DOI:** 10.3390/pathogens9030215

**Published:** 2020-03-13

**Authors:** Muhammad Akmal, Aryan Rahimi-Midani, Muhammad Hafeez-ur-Rehman, Ali Hussain, Tae-Jin Choi

**Affiliations:** 1Department of Fisheries and Aquaculture, University of Veterinary and Animal Sciences, Lahore 54000, Pakistan; muhammad.akmal@uvas.edu.pk (M.A.); mhafeezurehman@uvas.edu.pk (M.H.-u.-R.); 2Department of Microbiology, Pukyong National University, Busan 48513, Korea; aryan_rahimi2011@yahoo.com; 3Department of Wildlife and Ecology, University of Veterinary and Animal Sciences, Lahore 54000, Pakistan; ali.hussain@uvas.edu.pk

**Keywords:** *Aeromonas hydrophila*, aquaculture, bacteriophage, phage therapy, phage genomics

## Abstract

Bacteriophages are increasingly being used as biological control agents against pathogenic bacteria. In the present study, we isolate and characterize bacteriophage Akh-2 from Geoje Island, South Korea, to evaluate its utility in controlling motile *Aeromonas* septicemia. Akh-2 lysed four of the seven *Aeromonas hydrophila* strains tested. Transmission electron microscopy analysis showed that Akh-2 belongs to the Siphoviridae family, with head and tail sizes of 50 ± 5 and 170 ± 5 nm, respectively. One-step growth curve analysis revealed that the phage has a latent period of 50 ± 5 min and a burst size of 139 ± 5 plaque-forming units per infected cell. The phage appeared stable in a pH range of 6–8 and a temperature range of −80 to 46 °C. Based on next-generation sequencing analysis, its genome is 114,901 bp in size, with a 44.22% G + C content and 254 open reading frames. During an artificial induction of the disease, loach (*Misgurnus anguillicaudatus)* treated with Akh-2 showed an increased survival rate and time compared with the non-treated control. Our results suggest that Akh-2 is a potential biological agent for the treatment of *Aeromonas* infections in fish.

## 1. Introduction

Aquaculture is the fastest-growing food-production sector worldwide. In 2014, aquaculture produced nearly 74 million tons of fish, approximately 45% of the global production of fish-based food [1]. Due to the increasing demand for food protein and the stagnation or reduction of wild catch, fish production by aquaculture is expanding, with a growth rate exceeding 8% per year [1]. However, the increase in aquaculture has resulted in several problems, including the destruction of natural ecosystems, water pollution, biological contamination, and the emergence of diverse fish diseases [2]. Bacterial fish diseases are usually treated with antibiotics, and it is the reason for leakage to the environment and selection of resistance.

*Aeromonas* spp. are the most common bacteria in freshwater habitats and are frequently associated with severe infections in cultured fish species [3]. *Aeromonas hydrophila*, a Gram-negative rod-shaped bacterium, is the main cause of the disease motile *Aeromonas* septicemia, also known as tail and fin rot. This bacterium causes serious infections in various freshwater fish species, including loach (*Misgurnus anguillicaudatus*), channel catfish (*Ictalurus punctatus*), and common carp (*Cyprinus carpio*), and infects some marine fish species to a lesser extent [4]. Clinical signs of these infections include ulcers, abdominal distension, accumulation of fluid, anemia, and hemorrhaging, resulting in mass mortality in fishes around the world [4,5].

In farming practices, various methods such as vaccination, water chlorination, and antibiotic chemotherapy have been shown to reduce the occurrence of *Aeromonas* infection. However, excess use of antibiotics has resulted in the emergence of multidrug-resistant strains of *A. hydrophila* [6]. In this context, biological control seems to be a preferable alternative, and one proposed method is phage therapy. Phages are abundant in nature and can be used to lyse bacteria with the advantages of having high specificity toward their target and little to no negative impacts on the environment. Several studies have demonstrated the therapeutic effects of phages as substitutes for antibiotics in human and animal health problems [7,8].

Loach (*M. anguillicaudatus)* is widely cultured in South Korea due to its high demand as a food fish and its extensive use in traditional medicine and Buddhist ceremonies. *A. hydrophila* is one of the main causes of massive mortalities in loach, and the presence of multidrug-resistant *Aeromonas* further complicates its aquaculture [9]. Several studies have reported on the phage-based biocontrol of *Aeromonas* infection. However, issue with modern regulations and the slow process for getting phage therapy approval is the hurdle in field application, and the excessive use of antibiotics continues [6,10,11]. Although *A. hydrophila* is a major pathogen of freshwater fish, higher densities of *A. hydrophila* have been observed in saline habitats compared with freshwater habitats [12]. Considering the advantages of biological disease control in aquaculture and the prevalence of the host bacterium in seawater, the present study aims to isolate and characterize a potentially suitable bacteriophage against *A. hydrophila* from water samples collected from marine environments. The isolated phage was characterized and examined for its possible protective effects against experimentally induced motile *Aeromonas* septicemia.

## 2. Results

### 2.1. Isolation and Characterization of the Bacteriophage

From a total of 300 soil and water samples tested, two bacteriophages were isolated from one water sample collected from Wahyeon Beach using *A. hydrophila* (KCTC 2358) as a host. Plaques of different sizes were selected, and pure strains were established by three cycles of plaque isolation. Plaque size is an inherent characteristic of these two phages and showed little variation in each plaque isolation step. Therefore, we selected a phage designated as Akh-2, which produced bigger plaques for further characterization and application for phage therapy.

### 2.2. Specific Host Range and Morphology of Akh-2

The host range of phage Akh-2 was determined by spot assay. Among the 30 tested strains, Akh-2 lysed only four strains of *A. hydrophila* but could not lyse the other three strains of *A. hydrophila* or the remaining 23 tested bacteria (Table 1).

Transmission electron microscopy of Akh-2 revealed a phage with an icosahedral capsid of 50 ± 5 nm in diameter and a tail of 170 ± 5 nm in length (Figure 1), suggesting that Akh-2 belongs to the Siphoviridae family, which was confirmed by genome sequence analysis, as described below.

### 2.3. One-Step Growth Curve and Phage Stability

Based on the one-step growth curve (Figure 2), the latent period and burst size were calculated as ~50 ± 5 min and 139 ± 5 PFU/infected cell, respectively.

Phage maintained from −80 °C to 37 °C showed 99% infectivity after three days of incubation. When maintained at 45 and 50 °C for the same period, the phage titer was decreased by 15% ± 5% and 47% ± 5%, respectively. When kept at 55 or 60 °C, no active phage was observed after three days. The phage survival rate was 100% between 4 and 37 °C (Figure 3).

The phage was maintained from pH 4 to 11 for three days to test its pH stability. Phage maintained at pH 7 showed 100% survival, which did not decrease significantly at pH 8 or 9. Significant decreases in the phage titer were observed at pH 5 and 10 after three days; no phage infectivity was observed at pH 4 or 12 (Figure 4).

### 2.4. Genomic Analysis

The sequencing of Akh-2 generated 16,820,356 total reads and 1.070 Gbp with a G + C content of 45.22%. The size of the genome determined by de novo assembly using Platanus was 114,901 bp. A total of 254 ORFs were generated using ORF finder, of which 114 were encoded by the plus strand and 141 by the minus strand. Using BLAST and RAST, 187 ORFs were predicted to encode hypothetical proteins, 31 to encode tRNAs, and 36 to encode proteins with various functions (Appendix A). Most of the predictions corresponded to different structural proteins (i.e., tail fiber, major capsid, and phage protein), DNA ligases (repairing new DNA), and proteins involved in viral replication. No ORF encoding a gene involved the integration of viral genome into the bacterial genome that results in lysogenic infection was found. The genome was submitted to NCBI under accession number MK318083.1. Among the 17 *A. hydrophila* and *Aeromonas* phages of the Siphoviridae family listed by Bai et al. [13], phage Akh-2 showed the highest homology to AhSzw-1, with an overall nucleotide sequence identity of 74% (Figure 5).

### 2.5. Protective Effects of Akh-2

In the trial challenge test with *A. hydrophila*, loach immersed in bacterial solutions of 1 × 10^6^ (T1), 1 × 10^7^ (T2), and 1 × 10^8^ (T3) CFU/mL for 30 min showed 100% mortality within 72, 48, and 48 h, respectively. The bacterial dose of 1 × 10^7^ CFU/mL was selected for further trials.

The protective effect of phage Akh-2 was assessed over a 96-h period after infection. Group I, the control group treated with PBS, showed no mortality or disease signs during the experiment (Figure 6). Group II, the bacterial infection group challenged with *A. hydrophila* (1 × 10^7^ CFU/mL), showed sudden increases in mortality of 40% ± 1.52% and 96.33% ± 3.74% after 24 and 48 h of infection, respectively (Figure 6 and Figure 7A). A cumulative mortality rate of 100% with abdominal hemorrhaging and the presence of red spots on the body was observed within 72 to 96 h (Figure 7B). Group III, treated by immersion in water containing 1.0 × 10^8^ PFU/mL phage Akh-2 after challenge with *A. hydrophila*, showed cumulative mortality rates of 16% ± 3.60%, 53% ± 1.50%, 57% ± 3.60%, and 56.67% ± 3.78% after 24, 48, 72, and 96 h, respectively (Figure 6). Most of the surviving fish in this group showed no disease symptoms, but very small red spots could be observed on a few of the surviving fish by careful examination (Figure 7C). Group IV, treated with phage Akh-2 without bacterial infection, showed 100% survival with no symptoms, confirming the safety of the phage (data not shown). At the end of the experiment, *A. hydrophila* was re-isolated from the kidneys of dead or diseased fish, but not from the surviving phage-treated fish, confirming that the mortalities were caused by *A. hydrophila.*

## 3. Discussion

The appearance of antibiotic-resistant pathogenic bacteria is a severe problem in aquaculture, especially when there are few treatment options, such as in *A. hydrophila* infection of cyprinid loach [6,9]. In some cases, 50% mortality was observed even with the use of antibiotics, indicating the presence of antibiotic-resistant strains of *Aeromonas* species [9]. Bacteriophages have been used for the control of bacterial infection since their discovery. Their therapeutic effects have been confirmed in animals, humans, crops, and aquaculture [7,8,10]. Several studies have reported the isolation of *Aeromonas* phages and examined their protective effects against disease [3,14,15].

In the present study, we isolated bacteriophage Akh-2 infecting *A. hydrophila* (KTCT 2358) from seawater collected from Geoje Island, South Korea, and analyzed its potential as a phage therapy agent. Different phages that infect pathogenic bacteria of fish have been isolated from sea, riverine, and pond waters [16,17,18,19]. Although *A. hydrophila* is a major pathogen of freshwater fish, a high abundance of this bacterium in seawater has been reported [12]; thus, we attempted to isolate *A. hydrophila*-infecting phages from seawater samples. Yuan et al. [20] also reported the isolation of two *A. hydrophila*-infecting Siphoviridae members from seawater. Transmission electron microscopy suggested that Akh-2 belongs to the Siphoviridae family due to its icosahedral head and long non-contractile tail. Among the 60 viruses that infect the *Aeromonas* species, including nine novel phages reported recently [13], only six belong to the Siphoviridae family, and only 3 (AhSzq-1, AhSzw-1, and 4L372X) infect *A. hydrophila* [13,20]. Although possible applications of these viruses in fish disease control have been suggested, they have not been tested. However, different studies have reported that bacteriophages from other families can be used for the control of fish disease caused by *A. hydrophila* [3], and *Siphoviridae* family phages can be used for the treatment of *Pseudomonas plecoglossicida* infections in fish [21]. Therefore, we conducted molecular characterization of a Siphoviridae family phage that infects *A. hydrophila* and examined the therapeutic application of this phage.

Most bacteriophages are very specific for the receptors required for attachment and penetration of the host cell [18]. In the Akh-2 host range analysis, it infected only four of the seven tested *Aeromonas* strains, and no lytic effect was observed against the other non-*Aeromonas* species. Similar specific host range results were described by Kim et al. [22]. The high specificity of Akh-2 can be considered beneficial in terms of its effect on the healthy flora of aquatic organisms.

The one-step growth curve describes each of the different stages involved in the multiplication of a bacteriophage. For every environmental condition, there is a strong relationship between the latent period and burst size; that is, a high level of phage fitness is associated with an optimal latent time [23]. Phage Akh-2 has a latent period of 50 ± 5 min, which is almost the same as those of phages AhSzq-1 (50 min) and AhSzw-1 (60 min). Meanwhile, the burst size of Akh-2 was ~139 ± 5 PFU/infected cell, which is higher than that of phages AhSzq-1 and AhSzw-1 (45 PFU/infected cell) [20]. The burst size of a phage mainly depends on its plaque size and latent period, while the latent period can be affected by the type of phage, the host, and the environmental conditions [24]. The short latent period combined with the higher burst size makes Akh-2 a suitable candidate for phage therapy and could be used in combination with other phages as a cocktail for better control effects.

Many factors can affect the normal processes of phage infection, mainly attachment, penetration, and multiplication, but temperature and pH also play vital roles [25]. During the farming of warm water fish species, the optimal temperature range is 25 to 37 °C. The survival rate of phage Akh-2 was 100% between –80 °C and 37 °C, making it more suitable for long-term preservation and farm trails (Figure 2). With the highest survival rate observed at pH 7, Akh-2 shows reasonable stability around neutral pH, which is the optimum range for warm water fish farming.

The genome size (114,901 bp) and G + C content (45.22%) of Akh-2 are similar to those of *A. hydrophila* phages AhSzq-1 (112,558 bp and 43.86%, respectively) and AhSzw-1 (115,739 bp and 43.82%, respectively). AhSzw-1 encodes 32 predicted tRNA genes, whereas the genome of Akh-2 contains 31 tRNA genes. The presence of large numbers of tRNA genes in *A. hydrophila* phages is presumed to complement the codon usage bias of the host, which has a G + C content of 61% and more frequent usage of G or C at the third position [20].

In the genome of AKh-2, no ORF encoding a protein involved in the integration of viral genome such as an integrase was found. Temperate or lysogenic phages are generally not recommended for phage therapy [7]. The lysogenic phages can grant immunity against the same or related phages to the host. Also, the host bacteria can acquire new genetic traits such as phage-encoded toxins and resistance to antibiotic resistance determinants by phage conversion [7,17,19].

Several studies have reported the isolation of *Aeromonas* phages and the investigation of their protective effects against *Aeromonas salmonicida* [26,27,28] and *A. hydrophila* [3,14]. Although the results of these studies suggested their protective effects against *Aeromonas* infection, the degrees of protection are different. The methods of bacteria inoculation and phage administration, such as intraperitoneal injection [3], intramuscular injection [26], addition to water [28], and feeding phage impregnated feed [3], can affect the degree of protection. For example, in a phage therapy experiment against *A. hydrophila* in cyprinid loach (*M. anguillicaudatus*), the same fish species used in this experiment, the fish were infected by intraperitoneal injection with *A. hydrophila* at 2.6 × 10^7^ CFU/fish, causing 100% cumulative mortality in 7 days, and were treated by injection of phage or feeding with pellets that had been impregnated with phage suspensions [3]. In the case of intraperitoneal injection with phage pAh1-C (3.0 × 10^7^ PFU/fish) or pAh6-C (1.7 × 10^7^ PFU/fish), the cumulative mortalities were 43.33% ± 2.89% and 16.67% ± 3.82%, respectively. When the fish were given phage-impregnated feed, the cumulative mortality rates were 46.67% ± 3.82% (pAh1-C) and 26.67% ± 2.89% (pAh6-C), which indicated that injection provided better protection than phage feeding. In our experiment, we used the Af-clipping methods developed by Zhang et al. [29], which is more similar to the natural infection route. Immersion of Af-clipped fish for 30 min in water containing 1 × 10^7^ CFU/mL *A. hydrophila* resulted in 100% cumulative mortality at 48 h. Other factors such as fish species, bacterial strain, the order of bacteria inoculation, and phage administration and environmental conditions can also affect the degree of protection. One factor that must be considered is the MOI. MOIs of 6.5, 100, and 10,000 were used by Kim et al. [26], Silva et al. [27], and Imbeault et al. [28], respectively, and we used an MOI of 10. Due to multiplication and the resulting increase in phage titer in the presence of the host bacterium, the precise initial dose of phage has not been considered critical. In fact, Verner-Jeffreys [30] reported that phage therapy with an MOI of 100,000 was not effective for controlling *A. salmonicida.*

One consideration in phage therapy is the appearance of phage-resistant host bacteria. Although the exact mechanism of resistance, such as resistance by the CRISPR-CAS system, has not been determined, the occurrence of phage-resistant mutants has been reported in phage therapy against *Aeromonas* [3,27,31]. It has been suggested that phage resistance can be overcome by the phage due to the co-evolution of the phage and host [27]. Moreover, the slow growth of phage-resistant strains has been observed, which can limit the bacterial population to a level controllable by the host immune system [27,32]. One possible method to overcome the presence or occurrence of phage-resistant strains is the application of a phage cocktail rather than a single phage isolate, as shown by Chen et al. [33]. Although we isolated only two phages that can infect *A. hydrophila* from seawater, isolation of more *A. hydrophila* phages is promising, as shown by Hassan et al. [34], and further isolation, characterization, and application of *A. hydrophila* phages are necessary.

## 4. Materials and Methods

### 4.1. Isolation, Culture, and Characterization of the Bacteriophage

*A. hydrophila* strain KCTC 2358, which has been reported as a fish pathogen, was obtained from the Korean Collection for Type Cultures and used as a host bacterium for phage isolation. A total of 300 samples of seawater near a beach and soil from a beach were collected from nine locations on Geoje Island, South Korea. For phage isolation, 10 g of soil was mixed with 10 mL of sterilized distilled water and agitated 20 m at room temperature. The mixture or collected water samples were centrifuged at 3000× *g* for 20 min at 4 °C, and the resulting supernatant was filtered using a 0.2-µm filter (GVS Filter Technology, Indianapolis, Indiana). For phage enrichment, 100 µL filtrate, 4 mL nutrient broth, and 300 µL *A. hydrophila* overnight culture were mixed and incubated at 37 °C for 24 h. The cultures were centrifuged (3000× *g* for 20 min), and the supernatants were filtered as above. Then, an overlay mixture of 0.5 µL of the filtrate, 300 µL *A. hydrophila* overnight culture, and molten nutrient agar (0.7% agar) was poured over solidified nutrient agar (1.5%). After incubation at 37 °C for 24 h, clear phage plaques were picked using sterile end-cut pipette tips and stored at 4 °C in 1 M SM buffer (100 mM NaCl, 8 mM MgSO_4_·7H_2_O, and 50 mM Tris-Cl (pH 7.5)). Phage purification was performed as reported by Gencay et al. [35]. Phage titers were determined by the double-layer agar method. For long-term preservation, phages were preserved in 1 M SM buffer and stored at −80 °C.

### 4.2. Specific Host Range and Morphology of the Bacteriophage

A total of 30 strains, including seven strains of *A. hydrophila* (Table 1), were used for spot tests to confirm the host range of the phage. Purified phage particles (10^9^ plaque-forming units [PFU]/mL) preserved in SM buffer were adsorbed onto a carbon-coated copper grid and negatively stained with 2% uranyl acetate. After drying for 20 min at room temperature, the grids were observed using the JEM1010 electron microscope (Jeol, Tokyo, Japan) at the National Instrumentation Center for Environmental Management (NICEM) of Seoul National University, South Korea.

### 4.3. One-Step Growth Curve and Stability Analyses

One-step growth curve analysis was conducted according to Verma et al. [36] with some modifications. Briefly, 11 mL phage suspension (10^9^ PFU/mL) was added to a 20-mL overnight culture of the host bacterium for a multiplicity of infection (MOI) of 0.001, and the mixture was incubated at 37 °C for 5 min. After centrifugation at 35,000× *g* for 30 min, the supernatant was carefully removed, and the pellet was suspended in 20 mL nutrient broth. The mixture was incubated at 37 °C for 90 min, and samples were collected at 5-min intervals. The aliquots were diluted, and phage titers were determined by the double-layer agar method.

Phage stability was determined, according to Verma et al. [36], with slight modifications. Briefly, 1-mL aliquots of phage (10^7^ PFU/mL) were stored at various temperatures (−80, −20, 4, 25, 37, 45, 50, and 55 °C) in Eppendorf tubes for three days, and samples were collected every 24 h to determine the titer by the double-layer agar method. For the pH stability test, Tris-Cl was adjusted from pH 4 to 12. At each pH level, the phage was inoculated to a final concentration of 10^7^ PFU/mL and incubated at 37 °C for three days. Samples were taken every 24 h to determine the titer by the double-layer agar method.

### 4.4. Genomic Analysis

#### 4.4.1. Phage Genome Extraction

For nucleic acid isolation, 1 mL purified phage (10^9^ PFU/mL) was mixed with 0.5 µL DNase (1 U/µL) and 1 µL RNase A (10 mg/mL) and incubated for 1 h at 37 °C to remove any bacterial DNA contamination. Next, 20 µL EDTA (0.5 M), 1.25 µL proteinase K (20 mg/mL), and 100 µL 10% SDS were added to the phage solution, followed by a further incubation at 60°C for 1 h. After incubation, a 500-µL mixture of phenol, chloroform, and isoamyl alcohol (25:24:1) was added to the solution and mixed, followed by centrifugation at 12,000× *g* for 10 min. The supernatant was transferred to a new Eppendorf tube with a 1/10 volume of 3 M sodium acetate and a 2× volume of 100% isopropanol for nucleic acid precipitation, followed by incubation at −20 °C for 20 min and centrifugation at 12,000× *g* and 4 °C for 15 min. The nucleic acid pellet was washed with 70% ethanol, dried at room temperature, dissolved in 30 µL TE buffer, and stored at 4 °C until further use. The obtained genome was dissolved in distilled water, treated with DNase, RNase, and exonuclease II, and subjected to electrophoresis on an agarose gel to determine the property of the genome.

#### 4.4.2. Whole-Genome Sequencing, Assembly, and Annotation

Whole-genome sequencing was conducted using next-generation sequencing technology on the Illumina Hiseq sequencer at the Theragen Etex Bio Institute (Suwon, South Korea). The genomic DNA sample from the isolated phage was processed further for library preparation under the sample ID V2 TN1809D0396. The sequence reads were assembled de novo using Platanus version 1.2.2. Open reading frames (ORFs) were predicted using GeneMark.hmm, and their functions were annotated using the National Center for Biotechnology Information (NCBI) BLAST server and Rapid Annotations using Subsystems Technology (RAST) [37]. The genomes of Akh-2 and Ahszw-1 were compared by using Easyfig [38].

### 4.5. Protective Effects of Akh-2 in Infected Loach Fish

#### 4.5.1. Challenge Test and Estimation of the Lethal Dose

To simulate a more natural infection route similar to that in the aquatic environment, the methodology described by Zhang et al. [29] was adopted. A total of 120 healthy adipose fin-clipped (Af-clipped) loach with an average body weight of 10 g was distributed into four groups (each group consisting of three replicates of 10 fish) in four 30-L tanks filled with 15 L tap water that had been aerated overnight, which were labeled as control, T1, T2, and T3.

*A. hydrophila* (KCTC 2358) was cultured in NA medium, harvested by centrifugation at 3000× *g* for 20 min, resuspended in saline, and counted by plating. To determine the optimal challenge dose, T1, T2, and T3 groups were inoculated with final concentrations of 1 × 10^6^, 1 × 10^7^, and 1 × 10^8^ colony forming units (CFU)/mL, respectively. No bacteria were inoculated in the control group.

After a 30-min immersion with the respective treatment, fish were transferred to the experimental tanks with tap water and observed for 96 h for determination of clinical signs and mortality.

#### 4.5.2. Phage Treatment of Infected Fish

A total of 135 healthy loach with an average body weight of 10 g were divided into three groups (each consisting of three replicates of 15 fish) and placed in 30-L tanks filled with 15 L water. Af-clipping and the immersion method described above were used for the inoculation of bacteria and phage.

Group I was the control group in which loach were immersed in phosphate-buffered saline (PBS) for 30 min. Group II was the bacterial challenge group in which loach were immersed in water containing *A. hydrophila* at a concentration of 1 × 10^7^ CFU/mL for 30 min, as described above. Group III was the phage-treated group in which loach were immersed in water containing *A. hydrophila* (1 × 10^7^ CFU/mL) for 30 min, and then immediately immersed in water containing phage Akh-2 at a final concentration of 1 × 10^8^ PFU/mL for 30 min, maintaining an MOI of 10. In Group IV, loach were immersed for 30 min in water containing phage Akh-2 (1.0 × 10^8^ PFU/mL) without bacterial infection. After each immersion, loach were raised in separate 15-L water-filled experimental tanks and observed for 96 h without water exchange. Continuous aeration was provided, but food was not provided to maintain the water quality. The experiments were conducted three times on different occasions. The water was maintained at 28 ± 2 °C and pH 7 throughout the experimental trails. At the end of experiments, kidneys of dead or diseased fish were taken, ground in PBS and spread on LB plates after dilution. After the culture of grown bacterial colonies in LB medium, DNA was extracted and used as a template for PCR with universal primers for bacterial 16S rRNA gene, 27F (5’-AGAGTTTGATCCTGGCTCAG-3’) and 1492R (5’-CGGTTACCTTGTTACGACTT-3’). The PCR products were sequenced for the identification of bacteria.

### 4.6. Statistical Analysis

Statistically significant differences in all experiments were determined by Student’s *t*-test. A *p*-value <0.05 was considered to indicate statistical significance. The SPSS statistical software package (version 13.0) was used for all analyses.

### 4.7. Animal Experiment Approval

The animal protocol used in this study was approved by the Pukyong National University Institutional Animal Care and Use Committee (Approval Number PKNU-2018-12).

## Figures and Tables

**Figure 1 pathogens-09-00215-f001:**
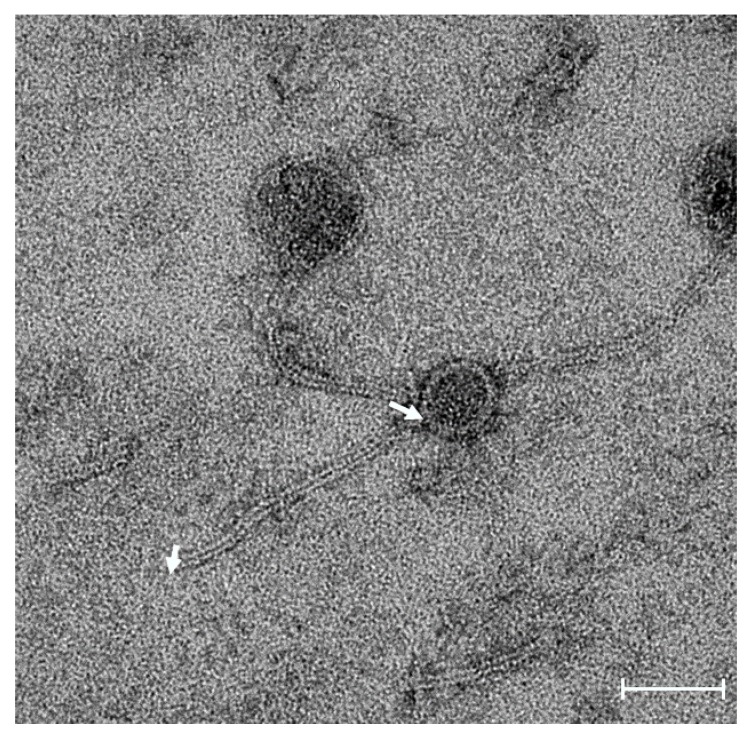
Electron micrograph of bacteriophage Akh-2. The two arrows indicate the beginning and end of the 170-nm-long tail. Scale bar = 50 nm.

**Figure 2 pathogens-09-00215-f002:**
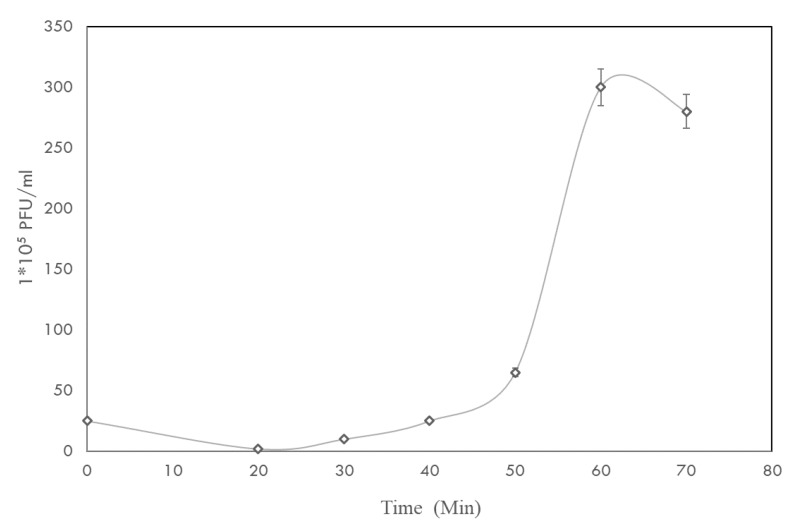
One-step growth curve of phage Akh-2. The results are the average of three replications with standard deviation as vertical lines.

**Figure 3 pathogens-09-00215-f003:**
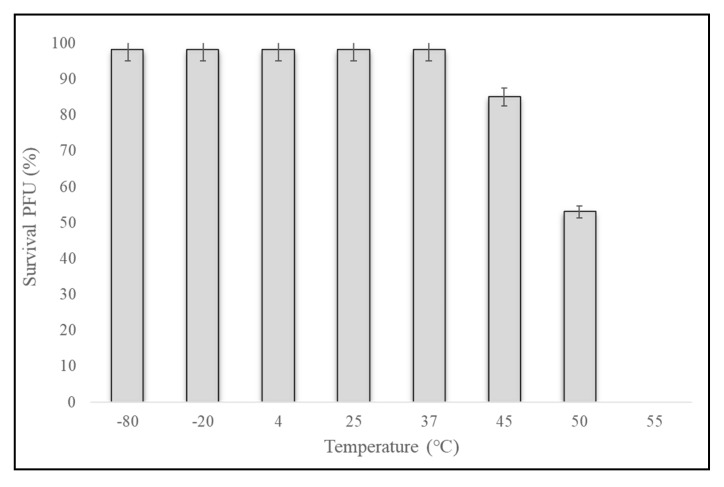
Stability of phage Akh-2 at different temperatures. Phage Akh-2 (10^7^ PFU/mL) was maintained at the indicated temperatures for three days, and then the titer was determined by plaque assay. The results are the average of three replications with standard deviation as vertical lines.

**Figure 4 pathogens-09-00215-f004:**
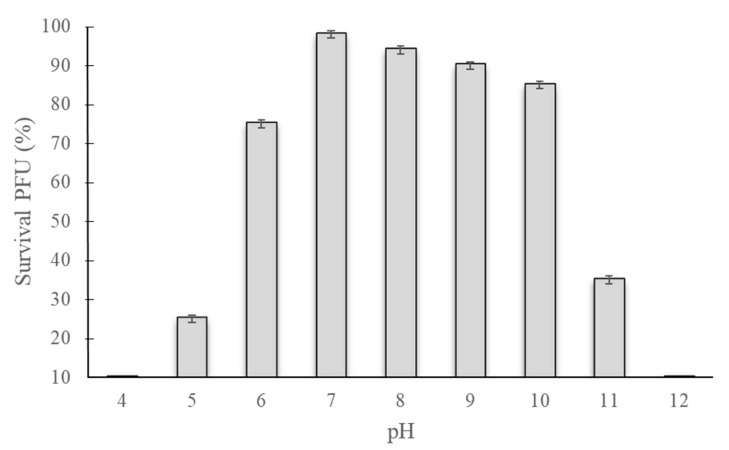
Stability of phage Akh-2 at different pH levels. Phage Akh-2 (10^7^ PFU/mL) was maintained at the indicated pH levels for three days, and then the titer was determined by plaque assay. The results are the average of three replications with standard deviation as vertical lines.

**Figure 5 pathogens-09-00215-f005:**
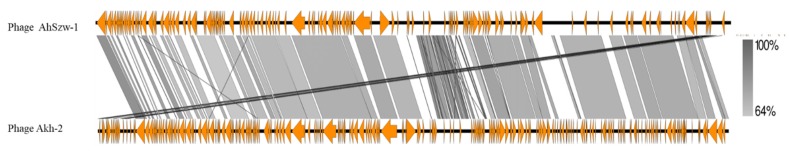
Genomic comparison of phage Akh-2 and the reference *A. hydrophila* phage AhSzw-1 (GenBank No. MG676225.1), constructed using EasyFigure. Arrows represent ORFs. The level of identity is indicated by the gray shading.

**Figure 6 pathogens-09-00215-f006:**
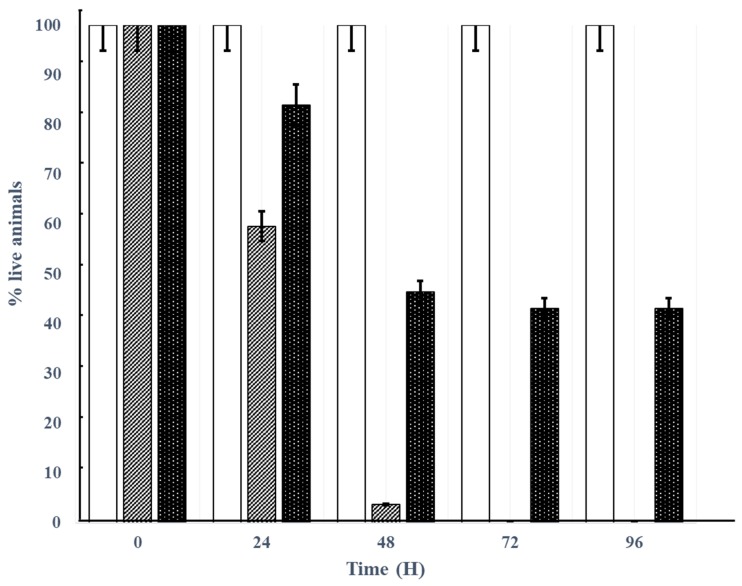
Protective effect of Akh-2 against *A. hydrophila* in inoculated loach. Af-clipped loach were immersed in PBS (□), *A. hydrophila* (1 × 10^7^ CFU/mL) (▨), or *A. hydrophila* (1 × 10^7^ CFU/mL), followed by Akh-2 (1 × 10^8^ PFU/mL) (■), and the survival rate was measured within 96 h. The results are the average of three replications with standard deviation as vertical lines (Appendix A).

**Figure 7 pathogens-09-00215-f007:**
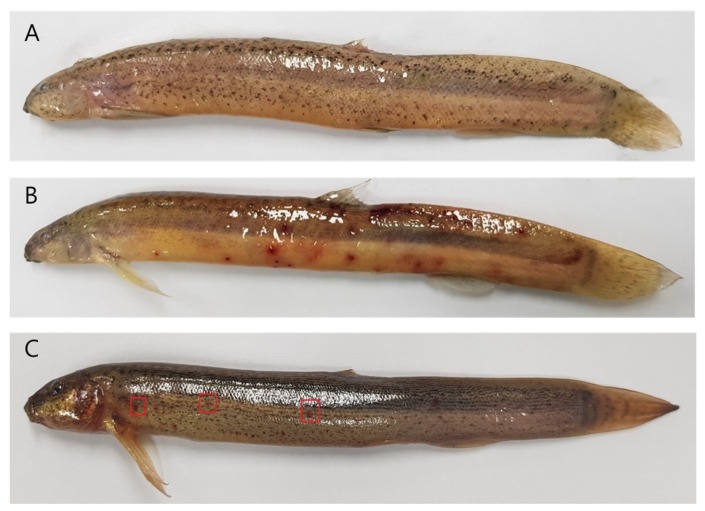
Protection of loach (*M. anguillicaudatus)* from *A. hydrophila* infection by bacteriophage Akh-2. (**A**) Non-infected negative control. (**B**) Loach inoculated with *A. hydrophila* showing hemorrhagic red spots. (**C**) Loach treated with phage Akh-2 after inoculation with *A. hydrophila* showing fewer and smaller hemorrhagic red spots (indicated by red boxes). The average fish size was 5 ± 2 cm.

**Table 1 pathogens-09-00215-t001:** Host range of Akh-2.

Bacteria	Strain ID	Lysis by Akh-2	Source *
*Aeromonas hydrophila*	KTCC 2358	YES	1
	KCCM 32586	NO	2
	AH-A6	YES	3
	AH-A8	NO	3
	AH-20	YES	3
	AH-21	NO	3
	AH-Juwah	YES	3
*Pectobacterium* spp.	35, 48, 63, 92, E42, E44	NO	4
*Streptococcus aureus*	S75, S86, S103, S106	NO	4
*Bacillus* spp.	B1, B2, B3, B4, B5, B6, B8, B11, B12, B13, B87, B97	NO	4
*Escherichia coli*	CJY H7	NO	4

* 1: Korean Collection for Type Cultures, 2: Korean Culture Center of Microorganisms, 3: Seoul National University, 4: Microbial Safety Division, National Institute of Agricultural Science, Rural Development Administration, Wanju-gun, Jeollabuk-do, Republic of Korea.

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
