# Peer review of "Isolation, Characterization, and Application of a Bacteriophage Infecting the Fish Pathogen Aeromonas hydrophila"

_pathogens, 2020, doi:10.3390/pathogens9030215_

Round 1
Reviewer 1 Report
Line 17: after "- 80"inserts "°C";
Line 29: before "destruction" inserts "the"
Line 71: substitutes "3" with "three" and "4" with "four"
Line 87: substitutes "3" with "three"
Line 89: substitutes "maintained" with "kept" and "3" with "three"
Line 91: substitutes "3" with "three"
Line 93: substitutes "3" with "three"
Line 100: substitutes "3" with "three"
Line 104: substitutes "3" with "three"
Line 133: substitutes "trials" with "tests"
Line 158: remove "an" before "agriculture"
Line 159: substitutes "serious" with "severe"
Line 172: write "host-pathogen"
Line 172: write "pathogenic fish bacteria"
Line 179: replace "9" and "6" with "nine" and "six"
Line 182: replace "other" with "different"
Line 191: replace "normal" with "healthy"
Line 224: insert "of" before "their"
Line 235: after "Many factors" add "," and change "environment" with "environmental"
Line 258: replace "experiments" with "trials"
Line 272: replace "is" with "are"
Line 292: write "plaque-forming"
Lines 307 and 310: change "3" with "three"
Line 333: insert "the" before "construction"
Line 339: change "were" with "was" and "change "labeled" with "labelled"
Line 361: change "separate" with "different"
Author Response
Dear Reviewer #1
We so much appreciate your kind detail review. We did our best to answer to all of your comments. All correction made to response are highlighted in red with corrections for reviewer 2 in blue. Because reviewer 2 requested shortening of the discussion, some of corrections for your comments might be removed. We tried our best to improve the manuscript by answering your comments but willing to consider again if you are not satisfied.
All corrections are listed below.
Line 17: after "- 80"inserts "°C";
=> "°C" was added
Line 29: before "destruction" inserts "the"
=> "the" was added
Line 71: substitutes "3" with "three" and "4" with "four"
=> replaced with “three” and “four”
Line 87: substitutes "3" with "three"
=> replaced with “three”
Line 89: substitutes "maintained" with "kept" and "3" with "three"
=> replaced with "kept" and "three"
Line 91: substitutes "3" with "three"
=> replaced with “three”
Line 93: substitutes "3" with "three"
=> replaced with “three”
Line 100: substitutes "3" with "three"
=> replaced with “three”
Line 104: substitutes "3" with "three"
=> replaced with “three”
Line 133: substitutes "trials" with "tests"
=> replaced with "tests"
Line 158: remove "an" before "agriculture"
=> “an” is removed
Line 159: substitutes "serious" with "severe"
=> substituted with “severe”
Line 172: write "host-pathogen"
=> corrected as "host-pathogen"
Line 172: write "pathogenic fish bacteria"
=> changed as "pathogenic fish bacteria"
Line 179: replace "9" and "6" with "nine" and "six"
=> replaced with “nine” and “six”
Line 182: replace "other" with "different"
=> replaced with "different"
Line 191: replace "normal" with "healthy"
=> replaced with "healthy"
Line 224: insert "of" before "their"
=> "of" was inserted
Line 235: after "Many factors" add "," and change "environment" with environmental"
=> "," was added and "environment" was replaced with "environmental"
Line 258: replace "experiments" with "trials"
=> replaced with "trials"
Line 272: replace "is" with "are"
=> with "are"
Line 292: write "plaque-forming"
=> changed as "plaque-forming"
Lines 307 and 310: change "3" with "three"
=> replaced with “three”
Line 333: insert "the" before "construction"
=> “the” was inserted
Line 339: change "were" with "was" and "change "labeled" with "labelled"
=> changed as “was” and “labelled”
Line 361: change "separate" with "different
=> changed as “different”

Reviewer 2 Report
In this manuscript the authors describe the isolation of two phages capable of infecting Aeromonas hydrophyla. Only one was characterized, from basic experiments to the efficacy of protecting animals in vivo. Overall, it is an interesting paper and its English language use is adequate. In a time in which antibacterial measures are needed, such manuscript can be of interest to the scientific community and adds up to the literature regarding phage therapy. I got a positive impression from the presented data, but as it is some revisions are needed.
I have raised comments and suggestions to the authors. These can be read below.
1) Introduction
1.1) Lines 31-32: It is clear to someone in the field that fish diseases can lead to antibiotic-related problems. However, it would be better if you make a more direct connection here, for the sake of the general readership. Briefly mention that bacterial fish diseases are treated with antibiotics, and it is the reason for leakage to the environment and selection of resistance.
1.2) Lines 53-54: Thinking again on a broader readership, please expand on the idea behind the contrast of successful phage therapy trials and lack of phage use on the field. Make clearer the issue with modern regulations and the slow process for getting phage therapy approved.
2) Results
2.1) Line 63: 300 samples were used and two viruses found. Please give (briefly) more details on the samples.
2.2) Line 65-66: please specify the number of plaque purifications instead of only mentioning “repeated”.
2.3) Line 68: So no additional experiments besides isolation and EM were made on phage Akh-10? Since you mention this other phage, and tell it was different from Akh-2, please provide an EM picture in figure 1 to illustrate your claim. And since you mention EM in line 68, maybe you could already mention figure 1 here.
2.4) Figure 1: First of all, provide more pictures if possible, at different magnifications and different fields of view, to better illustrate the morphology. Secondly, I understood that one arrow is pointing to the beginning of the tail of one virion and the other arrow is pointing to the end of the tail of another virion. Is that right? If so, make it clearer. You could for example mark the beginning and end of the same tail.
2.5) Figure 2: Adding arrows to mark the latent period and the burst may not be necessary.
2.6) Lines 88-89: Since you had triplicates, please indicate the standard deviation of your percentages.
2.7) Figure legends (overall): please standardize how you mention your number of replicates and SD in each legend.
2.8) Line 114: I could not retrieve your phage genome from NCBI. Is it not available yet?
2.9) Line 115: Error in the citation format. Add the reference number, not year of publication.
2.10) Lines 114- 119 and figure 6: Is the DNApol comparison really necessary? Sequence identity of the whole genome, already mentioned and shown in Fig5, is more robust than just the polymerase.
2.11) Line 133-146 and figures 7-8: There is no mention of Fig7 here. You could join Fig7 and Fig8, having the pictures as the first items and the graph as the last. On line 140-141 you mention Figure 5, and it makes no sense. This paragraph should be checked for consistency.
2.12) Line 138: You should correct the formatting in “Fig 8B”, since you write “Figure” in other figure mentions.
2.13) Figure 7: Please consider showing this data in the classical survival graph style (% live animals x time) if possible.
3) Discussion
3.1) Overall: The discussion is too long and contains many speculations that could be removed. If relevant, check the text and shorten it.
3.2) Lines 158-160: Confusing sentence. Please rewrite.
3.3) Lines 168-169: This sentence gives a biased idea. There are large amounts of phages in the ocean indeed. But these are mostly phages that infect ocean protists, the majority unrelated to disease in animals. Please rewrite or remove.
3.4) Lines 203-204: This is only speculative. Latent period and burst size could mean nothing if bacteria is capable of quickly becoming resistant to your phage. You have even shown that protection using only this phage is not 100%, despite the moi of 10 tested. Your phage has potential for phage therapy, but for optimal protection, it could be one of a few in a cocktail for example.
3.5) Lines 211-223: Phages were shown to be able to bind to animal mucins, with relevance for finding hosts and for protection of the animal providing the mucosal layer. Mucus binding has even been shown to be important for phage therapy in a fish model (Flavobacterium columnare and rainbow trout). Since in your model Aeromonas causes skin disease, adherence to mucus might be relevant. Is it possible to check in silico for Ig-like domains in your phage structural proteins? If these are present, then you could theoretically link your phage to the bacteriophage adherence to mucus model. It could even be the first example of a siphovirus potentially behaving like the tailed phages used to test the mucus binding theory.
3.6) Lines 224-234: How these studies differ in relation to phage exposure timing (before, together or after the bacterial infection)? Can a trend be seen regarding phage timing and protection? Please mention here if relevant.
3.7) Line 243: The same fish species, and not the same fish.
4) Methods
4.1) Line 291: Another error in the citation format. Add the reference number, not year of publication.
4.2) Lines 309-310: did phage addition change the pH of the solutions?
4.3) Lines 323-325: why would you treat your extracted DNA with nucleases?
4.4) Line 346: what is normal water? How it differ from the water used previously?
4.5) Fish experiments: please provide more details, like feeding routine and mention aeration and water flowthrough (if used). Also, make a unique topic for your approval for fish experimentation. It is now in lines 362-364, but would be better if not mixed in the text.
5) References (overall): please reconsider reference use. Aim to cite the most relevant and original papers for the sentences you want to reference. For example: for referencing affirmations regarding Aeromonas diseases, you use two specific references (5,6) about phage therapy against Aeromonas. It would be better to cite classical Aeromonas disease papers instead if possible. The same with references in the methods section. The techniques used (isolation, titration, spot tests and so on) are common, and definitely not described for the first time in the papers cited. Please check all and change whenever necessary.
6) Supplementary information: The link provided in the manuscript is broken and I could not access it. So my revision is not taking the supplement into account.
Author Response
Please refer the attached file
